# 1α,25(OH)_2_D_3_ Promotes the Autophagy of Porcine Ovarian Granulosa Cells as a Protective Mechanism against ROS through the BNIP3/PINK1 Pathway

**DOI:** 10.3390/ijms24054364

**Published:** 2023-02-22

**Authors:** Shiyou Wang, Qichun Yao, Fan Zhao, Wenfei Cui, Christopher A. Price, Yifan Wang, Jing Lv, Hong Tang, Zhongliang Jiang

**Affiliations:** 1Key Laboratory of Animal Genetic, College of Animal Science and Technology, Breeding and Reproduction in Shaanxi Province, Northwestern A&F University, Xianyang 712100, China; 2Centre de Recherche en Reproduction et Fertilité, Faculté de Médecine Vétérinaire, Université de Montréal, St-Hyacinthe, QC J2S 7E4, Canada; 3State Key Laboratory for Sheep Genetic Improvement and Healthy Production/Institute of Animal Husbandry and Veterinary Medicine, Xinjiang Academy of Agricultural and Reclamation Sciences, Shihezi 832000, China

**Keywords:** 1α,25(OH)_2_D_3_, porcine granulosa cells, autophagy, ROS, signaling pathway

## Abstract

Vitamin D (VD) is one of the important nutrients required by livestock; however, VD deficiency is reported to be widespread. Earlier studies have suggested a potential role for VD in reproduction. Studies on the correlation between VD and sow reproduction are limited. The aim of the current study was aimed to determine the role of 1,25-dihydroxy vitamin D_3_ (1α,25(OH)_2_D_3_) on porcine ovarian granulosa cells (PGCs) in vitro to provide a theoretical basis for improving the reproductive efficiency of sows. We used chloroquine (autophagy inhibitor) and reactive oxygen species (ROS) scavenger N-acetylcysteine in conjunction with 1α,25(OH)_2_D_3_ to explore the effect on PGCs. The results showed that 10 nM of 1α,25(OH)_2_D_3_ increased PGC viability and ROS content. In addition, 1α,25(OH)_2_D_3_ induces PGC autophagy according to the gene transcription and protein expression levels of LC3, ATG7, BECN1, and SQSTM1 and promotes the generation of autophagosomes. 1α,25(OH)_2_D_3_-induced autophagy affects the synthesis of E_2_ and P_4_ in PGCs. We investigated the relationship between ROS and autophagy, and the results showed that 1α,25(OH)_2_D_3_-induced ROS promoted PGC autophagy. The ROS-BNIP3-PINK1 pathway was involved in PGC autophagy induced by 1α,25(OH)_2_D_3_. In conclusion, this study suggests that 1α,25(OH)_2_D_3_ promotes PGC autophagy as a protective mechanism against ROS via the BNIP3/PINK1 pathway.

## 1. Introduction

Ovarian granulosa cells (GCs) play a pivotal role in follicle growth and atresia. Autophagy is a process of self-phagocytosis widely present in almost all eukaryotes and is one of the degradation pathways of redundant or abnormal cellular components. The molecular pathways that regulate autophagy are highly conserved. As an important autophagy-related protein, ATG7 is associated with autophagosome formation. BECN1 (Beclin1) forms a complex with the class III phosphoinositol 3-kinase molecule Vps34, which initiates and promotes autophagy. Microtubule-associated protein 1 light-chain 3 (LC3) is essential for the formation and maturation of autophagosomes. SQSTM1(P62) protein functions as a selective autophagy receptor for the degradation of substrates. In the ovary, GC autophagy affects follicle development. Autophagy occurs in GCs of porcine follicles [1]. Previous studies showed that autophagy is the leading cause of follicular atresia in neonatal mice [2], and autophagy-related genes and proteins are continuously expressed during cytogenesis. ATG7 is expressed in oocytes, and LC3 exists in GCs [3]. Autophagy is induced specifically in GCs during folliculogenesis. The LC3 protein is expressed mainly in GCs during all developmental stages [4]. Autophagy is closely related to the growth, proliferation, and apoptosis of GCs [5]. Inadequate autophagy of GCs leads to reduced progesterone synthesis [6] and disruption of GC differentiation [7]. Our previous studies showed that mir-21-3p regulates autophagy in bovine granulosa cells through the PI3K/AKT signaling pathway [8]. Both FSH [9] and ERβ [10] can induce autophagy in bovine ovarian granulosa cells via AKT/mTOR pathway. The underlying mechanism of GC autophagy remains to be determined, and the relationship between GC autophagy and follicle development remains unclear.

Vitamin D (VD) is a steroid derivative with a wide range of biological properties, and its main active form is 1,25-dihydroxy vitamin D_3_ (1α,25(OH)_2_D_3_). Studies conducted over the past 20 years have found that VD plays a vital role in maintaining the regular female reproductive system. Vitamin D and its clinical implications regarding the developmental competence and fertilization of oocytes has prompted researchers’ attention. A recent study revealed that in the case of VD deficiency and hypocalcemia at the same time, a significant reduction in oocyte retrieval after ovarian stimulation was observed, and the generated oocytes showed a poor maturation ability [11]. Moreover, VD signaling leads to an increased production of steroid hormones in granulosa cells, which are crucial for oocyte maturation and pregnancy, too. Vitamin D is essential for female gametes and their micro-environment [12]. Vitamin D might improve follicular development and subsequently oocyte quality. The biological effects of VD are usually achieved through the vitamin D receptor (VDR), which is found in the female reproductive system, including the uterus, endometrium, ovary, and placenta [13]. Vitamin D stimulates the production of estrogen, progesterone, and IGF-binding protein 1 in human ovarian cells [14]. To date, most of the studies on VD in follicle development have focused on humans. The effect of VD on porcine follicle development has been less reported. The classic function of VD is to maintain musculoskeletal health by maintaining calcium homeostasis. Meanwhile, it has been shown that VD can affect the equilibrium state of oxidation/reduction in C2C12 cells [12], resulting in a balance of ROS production and elimination. Vitamin D affects the oxidative capacity of cells by regulating the activities of superoxide dismutase (SOD), catalase (CAT), and glutathione peroxidase (GPX) [15]. Vitamin D’s effects on the redox status of porcine ovarian granulosa cells has not been reported. In addition, VD plays an essential role in the autophagy induced by primary monocytes and macrophages [16]. Vitamin D was reported to regulate autophagy via calcium ions [17], the PI3K/AKT/mTOR pathway [18], inflammatory factors [19], antimicrobial peptide [16], etc. In recent years, increasing evidence has demonstrated the critical role of GC autophagy in follicle development and atresia [20]; however, the effect of VD on the autophagy of GCs remains unclear. In the current study, we assumed that 1α,25(OH)_2_D_3_ regulates the PGC autophagy and affects the redox status and function of PGCs. To test this hypothesis, we examined the effects of 1α,25(OH)_2_D_3_ on intracellular ROS content and autophagy in PGCs and evaluated the effects of 1α,25(OH)_2_D_3_ on 17β-estradiol (E_2_) and progesterone (P_4_) secretion in PGCs.

## 2. Results

### 2.1. Effects of 1α,25(OH)_2_D_3_ on the Viability of PGCs

We first investigated the effects of doses and times of 1α,25(OH)_2_D_3_ treatment on PGC viability and the expression of cell-cycle-related genes in PGCs (Figure 1). The proliferation of PGCs (treatment with 10 nM 1α,25(OH)_2_D_3_) was observed under a light microscope (Figure 1A). The results indicated that the viability was significantly increased in the groups of 10, 100, and 200 nM of 1α,25(OH)_2_D_3_ treatments for 12 h, 24 h, and 36 h in comparison to the control group, respectively (Figure 1B). The PGCs with the treatment of 10 nM of 1α,25(OH)_2_D_3_ for 24 h were selected for further studies. Then, quantitative real-time PCR was used to detect the expression of cell-cycle-related genes. The results showed that the gene expression of *VDR, CDK1* and *CCNB1* was increased, while the expression of *P21* was decreased in 1α,25(OH)_2_D_3_ treatment compared with the control (Figure 1C–F). These results suggest that 1α,25(OH)_2_D_3_ promotes the viability of PGCs.

### 2.2. 1α,25(OH)_2_D_3_ Increases Intracellular ROS in PGCs

To study the role of 1α,25(OH)_2_D_3_ on intracellular ROS in PGCs, PGCs were treated with 1α,25(OH)_2_D_3_, and the intracellular ROS was determined by DCFH-DA staining and observed under a fluorescence microscope (Figure 2A). The results indicated that 1α,25(OH)_2_D_3_ significantly increased the ROS content (Figure 2B). Similar results were confirmed by flow cytometry (Figure 2C,D). The results of relative gene expression in PGCs showed that 1α,25(OH)_2_D_3_ significantly down-regulated the expression of *SOD1* and *GSH-PX-1* genes in PGCs (Figure 2E,F); however, 1α,25(OH)_2_D_3_ did not change the *CAT* gene relative expression gene in PGCs (Figure 2G). Our results showed that VD_3_ significantly decreased enzyme activities of SOD (Figure 2H) and GPX (Figure 2I) in PGCs. Together, these results indicated that 1α,25(OH)_2_D_3_ increased intracellular ROS content in PGCs.

### 2.3. Mitochondria Status in the 1α,25(OH)_2_D_3_-Treated GCs

Mitochondria are the primary source of ROS in cells. To study the relationship between 1α,25(OH)_2_D_3_ and ROS, Mito-Tracker was used to label the PGC mitochondria in this experiment (Figure 3A). The results showed that 1α,25(OH)_2_D_3_ significantly increased the abundance of mitochondria in PGCs (Figure 3B). Moreover, 1α,25(OH)_2_D_3_ increased considerably the relative expression of the *ND1* gene, a mitochondrial DNA (mtDNA, Figure 3C). In contrast, the current results showed that 1α,25(OH)_2_D_3_ did not change the mitochondrial membrane potential of PGCs in this experiment (Figure 3D). These results suggest the mitochondria status in the 1α,25(OH)_2_D_3_-treated GCs.

### 2.4. 1α,25(OH)_2_D_3_ Induces PGC Autophagy

The cumulative data suggest that mitochondria play an important role in activating autophagy. To study the effects of 1α,25(OH)_2_D_3_ on PGC autophagy, MDC was used to label the autophagic vacuoles in PGCs treated by 1α,25(OH)_2_D_3_ and chloroquine (an inhibitor of autophagy) in this experiment, and the fluorescence of autophagic vacuoles in PGCs was detected by fluorescence microscope (Figure 4A). The current results showed that the treatments of 1α,25(OH)_2_D_3_, chloroquine significantly increased the number of autophagic vacuoles (Figure 4B). The relative expression of *ATG7*, *Beclin1*, and *LC3* genes were significantly up-regulated in PGCs treated with 1α,25(OH)_2_D_3_, chloroquine, and 1α,25(OH)_2_D_3_ with chloroquine, respectively (Figure 4C–E). The treatment of 1α,25(OH)_2_D_3_ down-regulated the *P62* mRNA expression, while chloroquine significantly up-regulated *P62* mRNA expression in the PGCs compared with that of the control group; however, the treatment of 1α,25(OH)_2_D_3_ with chloroquine did not change the *P62* mRNA expression in comparison to the control group (Figure 4F). Figure 4G showed the expression of autophagy proteins, and the results showed that the expression pattern of the P62 protein was similar to that of its gene in PGCs (Figure 4H). Moreover, 1α,25(OH)_2_D_3_ significantly increased the LC3II/LC3I level compared with that of the control group, while the treatments of chloroquine and 1α,25(OH)_2_D_3_ with chloroquine decreased the LC3II/LC3I level, respectively (Figure 4I). In addition, the LC3II/LC3I levels in the cells treated with chloroquine were lower than the treatment of 1α,25(OH)_2_D_3_ with chloroquine (Figure 4I). These results suggest that 1α,25(OH)_2_D_3_ induces PGC autophagy.

### 2.5. The Effects of 1α,25(OH)_2_D_3_ on Steroid Production of PGCs through Autophagy

To investigate the effects of 1α,25(OH)_2_D_3_ on steroid production, the PGCs were treated by 1α,25(OH)_2_D_3_, chloroquine and 1α,25(OH)_2_D_3_ with chloroquine together, and the contents of E_2_ and P_4_ were measured in this experiment. The results showed that compared with the control group, the concentration of E_2_ in PGC medium treated with 1α,25(OH)_2_D_3_ was significantly increased. In contrast, that in PGCs treated with chloroquine was significantly decreased (Figure 5A). Furthermore, the co-treatment of 1α,25(OH)_2_D_3_ and chloroquine did not change the E_2_ concentration in PGCs compared to the control group (Figure 5A). Although 1α,25(OH)_2_D_3_ increased the concentration of P_4_ in PGCs, both the treatments of chloroquine and 1α,25(OH)_2_D_3_ with chloroquine did not change P_4_ production in PGCs compared to the control group (Figure 5B). The treatments of 1α,25(OH)_2_D_3_ and 1α,25(OH)_2_D_3_ with chloroquine significantly up-regulated the relative expression of *ESR1, CYP19A1, PGR,* and *STAR* genes in PGCs in comparison to the control group; nevertheless, chloroquine did not change the expression of these genes (Figure 5C–F). The STAR protein level was identified by Western blotting (Figure 5G), and the results showed that 1α,25(OH)_2_D_3_ significantly increased the STAR level in PGCs in comparison to the control group; however, the treatments of chloroquine or 1α,25(OH)_2_D_3_ with chloroquine did not change the STAR level in PGCs (Figure 5H). These results suggest that the steroid production in PGCs was affected by 1α,25(OH)_2_D_3_-induced autophagy.

### 2.6. 1α,25(OH)_2_D_3_-Induced ROS Promotes Autophagy in PGCs

To explore the effects of 1α,25(OH)_2_D_3_-induced ROS on PGC autophagy, ROS scavenger N-acetylcysteine (NAC) was used to treat the PGCs for 24 h, followed by 1α,25(OH)_2_D_3_ treatment. The results indicated that NAC significantly decreased the viability of PGCs with or without 1α,25(OH)_2_D_3_ treatment (Figure 6A). As Figure 6B shown, ROS content was significantly reduced in NAC-treated PGCs, and the cells co-treated with 1α,25(OH)_2_D_3_ and NAC (4, 8 mM). Based on the results above, 4 mM NAC was used for the following experiments. DCFH-DA staining was used to detect the ROS, and similar results as those above were observed in PGCs treated with 4 mM NAC (Figure 6C,D). Moreover, 4 mM of NAC significantly decreased the number of autophagic vacuoles induced by 1α,25(OH)_2_D_3_ treatment in PGCs (Figure 6E,F). To confirm the results of MDC staining, the expression of autophagy-related genes in PGCs was measured. NAC down-regulated the 1α,25(OH)_2_D_3_-stimulated expression of *ATG7*, *Beclin1*, and *LC3* (Figure 6G–I) and up-regulated the *P62* mRNA expression (Figure 6J). In PGCs treated with NAC in the presence/absence of 1α,25(OH)_2_D_3_, LC3, and P62 protein expression patterns were observed to be similar to their gene expression patterns (Figure 6K–M). Together, we demonstrated that 1α,25(OH)_2_D_3_-induced ROS promotes autophagy in PGCs.

### 2.7. 1α,25(OH)_2_D_3_ Induces Mitophagy in PGCs through the ROS-BNIP3-PINK1 Signaling Pathway

To study the orientation of 1α,25(OH)_2_D_3_-induced autophagy, PGCs were treated with 1α,25(OH)_2_D_3_, chloroquine, and NAC. RT-qPCR was used to measure gene expression, and Western blotting was used to determine the protein levels in this experiment. Both 1α,25(OH)_2_D_3_ and chloroquine up-regulated the relative expression of *BNIP3* (the marker of mitophagy) and *PINK1* gene in PGCs compared with that of the control (Figure 7A,B). Furthermore, the protein levels of BNIP3 and PINK1 were increased in the treatments of 1α,25(OH)_2_D_3_, chloroquine, and 1α,25(OH)_2_D_3_ with chloroquine, which was similar to the relative expression of genes (Figure 7C–E). Compared with the treatment of 1α,25(OH)_2_D_3_, the relative expression of *BNIP3* and *PINK1* genes were significantly decreased in treatments of PGCs of NAC with 1α,25(OH)_2_D_3_ and NAC alone (Figure 7F,G). Moreover, the protein levels of BNIP3 and PINK1 were similar to the expression patterns of their genes in the treatments of 1α,25(OH)_2_D_3_, 1α,25(OH)_2_D_3_ with NAC, and NAC in PGCs (Figure 7H–J). We found that 1α,25(OH)_2_D_3_ induces mitophagy in PGCs through the ROS-BNIP3-PINK1 signaling pathway.

## 3. Discussion

1,25-dihydroxy vitamin D_3_ is a lipid-soluble secosteroid hormone established to play a wide range of biological functions [21]. More and more studies have shown that VD plays a vital role in life processes, including reproduction [22]. Breeding sows are the foundation of pig farm production, and their fecundity plays a crucial role in the benefit of the pig farms. Studies on the correlation between VD and sow reproduction are limited. Although studies have revealed the role of 1α,25(OH)_2_D_3_ on autophagy via the PI3K/AKT/mTOR pathway, the mechanism of 1α,25(OH)_2_D_3_ in the autophagy of ovarian granulosa cells remains unclear. Here, we demonstrate that (1) 1α,25(OH)_2_D_3_ increased porcine ovarian granulosa cell viability and the ROS content by increasing the number of mitochondria and decreasing the activities of superoxide dismutase and glutathione peroxidase; (2) porcine ovarian granulosa cell autophagy is regulated by 1α,25(OH)_2_D_3_ and affected the synthesis of E_2_ and P_4_; and (3) the ROS-BNIP3-PINK1 pathway was involved in porcine ovarian granulosa cell autophagy induced by 1α,25(OH)_2_D_3_.

The present results showed that 1α,25(OH)_2_D_3_ increased the *VDR* mRNA expression and promoted the proliferation of PGCs. This finding suggests that VD may play an important role in follicle development, which was supported by studies of VDR expression in goats [23] and mice [13]. Cell proliferation is controlled by the balance between cyclin-dependent kinases (CDKs) and its inhibitor (CKI). The expression of CDK1 and CCNB1 promotes the increase of cell number [24], and CDK function is tightly regulated by CKIs such as P21, which is related to cell cycle proliferation [25]. Our results showed that 1α,25(OH)_2_D_3_ up-regulates the expression of *CDK1* and *CCNB1* genes, while the expression of *P21* was down-regulated in PGCs. The mechanism by which 1α,25(OH)_2_D_3_ regulates GC proliferation remains incompletely understood. The mechanism by which 1α,25(OH)_2_D_3_ regulates the cell cycle process needs to be further studied.

In this study, the results of fluorescence microscope and flow cytometry analysis in PGCs indicated that 1α,25(OH)_2_D_3_ increased the content of ROS with down-regulation of *SOD1* and *GSH-PX-1* genes and the reduction of SOD and GSH enzyme activities. Generally, VD has antioxidant abilities through the reduction of ROS production to decrease oxidative stress [15]; however, VD also induces ROS production as a byproduct in reproductive tissues accompanied by steroidogenesis [26]. Antioxidant enzymes can eliminate excessive ROS production and protect the redox homeostasis in cells. Although VD deficiency has been associated with increased SOD enzyme activity in patients with chronic low back pain [27], a study showed that SOD enzyme activity was lower in rats deficient in VD. The absence of vitamin D leads to decreased SOD activity in vivo and in vitro. Vitamin D deficiency led to an increase in activities of the glutathione-dependent enzymes and a decrease in SOD and catalase enzymes in rat muscle [28]. These studies suggest that vitamin D supplementation is associated with changes in antioxidant enzyme activity. The current data indicates that 1α,25(OH)_2_D_3_ treatment significantly decreased SOD and GPX enzyme activities in PGCs.

Intracellular ROS mainly originate from mitochondria. Our results showed that 1α,25(OH)_2_D_3_ treatment increased the expression of the *ND1* gene, a mitochondrial metabolism-related gene, and the abundance of mitochondria in PGCs. Previous studies have demonstrated that VD is related to mitochondrial density [29] and the direct role of VDR in regulating mitochondrial respiration in skeletal muscle in vitro [30]. Mitochondrial abundance and mitochondrial DNA (mtDNA) copy number determine the metabolic activity of mitochondria [31]. The integrity of mtDNA and the activation of transcription and translation processes are essential for the induction of mitochondrial activity [32]. The mitochondrial mRNA transcription (such as mt-ND1~mt-ND6, CoxI~CoxIII) and their translation processes are activated with the increase of mitochondrial activity in serum-stimulated HeLa cells [33]. Meanwhile, the present results indicated that 1α,25(OH)_2_D_3_ did not affect the PGC mitochondrial membrane potential (MMP), which reflects the mitochondria functional status and is thought to be correlated with the cell differentiation status, tumorigenicity, and malignancy [34]. Mitochondrial fusion requires an intact MMP. The dissipation of MMP results in the rapid fragmentation of mitochondrial filaments, reforming interconnected mitochondria upon the withdrawal of MMP inhibitors [35]. The present results showed that 1α,25(OH)_2_D_3_ increased mitochondrial activity without negatively affecting mitochondrial function.

Our results demonstrate that 1α,25(OH)_2_D_3_ is responsible for autophagy in PGCs. Autophagy is influenced by various factors and environmental stimuli, including oxidative stress [36], starvation, and epigenetic regulation [37]. It has been reported that VD can regulate autophagy in different degrees, including induction, maturation, and degradation [38]. A particular concentration of 1α,25(OH)_2_D_3_ can induce autophagy in primary monocytes and macrophages [16]. Meanwhile, the association between vitamin D and autophagy has also been reported in immunity [39] and cancer [40]. 1α,25(OH)_2_D_3_ plays a protective role in acute myocardial infarction through autophagy induced by the PI3K/AKT/mTOR pathway [18]. Active vitamin D attenuates osteoarthritis by activating autophagy in chondrocytes through the AMPK-mTOR signaling pathway [41]. In addition, 1α,25(OH)_2_D_3_ can reduce cell dysfunction and intracellular oxidative stress by lowering excessive autophagy in cells [42]. These studies suggest that vitamin D can maintain cellular homeostasis by promoting or inhibiting autophagy. The current results showed that 1α,25(OH)_2_D_3_ promotes autophagy in PGCs, and the elevation of PGC autophagosome was confirmed by MDC staining.

GCs are one of the primary cell types in the follicle, and steroidogenesis is an essential physiological process affecting follicle maturation and ovulation. Here, we found that 1α,25(OH)_2_D_3_ promoted the secretion of E_2_ and P_4_ in PGCs. The current results showed that 1α,25(OH)_2_D_3_ stimulates the expression of *ESR1*, *PGR*, *CYP19A1*, and *STAR* mRNA in PGCs and increases the concentration of E_2_ and P_4_ in PGCs. The bilateral role between hormone secretion and autophagy has been confirmed in GCs of bovine [9] and goose follicles [43], which showed that hormone secretion induces autophagy; on the contrary, autophagy promotes hormone secretion in granulosa cells. In the present study, the results of co-treatments of 1α,25(OH)_2_D_3_ and chloroquine on PGCs reveal the weak expression of *ESR1*, *CYP19A1*, *PGR*, and *STAR* mRNA and lower STAR protein level in PGCs compared with that of 1α,25(OH)_2_D_3_ alone. These results suggest that 1α,25(OH)_2_D_3_-induced autophagy in PGCs promotes steroid hormone synthesis by regulating steroid synthesis enzymes. No reports exist about a possible role of autophagy in steroid hormone synthesis in GCs, but autophagy is implicated in the development and regression of ovarian cells. Autophagy is involved in the death of rat luteum cells through apoptosis, which is most evident in corpus luteum regression [44]. The accumulation of autophagosomes induces apoptosis of granulosa cells [3]. In addition, a link between steroid hormones and autophagy has been reported in farm animals. E_2_ and P_4_ increased autophagy in bovine mammary epithelial cells in vitro [45]. E_2_ and P_4_ may regulate mammary gland development, proliferation, and apoptosis of mammary epithelial cells in dairy cows by inducing autophagy [46].

Oxidative stress is one of the impact factors for cellular autophagy, and mitochondria are the primary source of intracellular ROS. In this study, the role of ROS in the autophagy of PGCs was investigated, and the results revealed that 1α,25(OH)_2_D_3_-induced ROS promotes the PGC autophagy. Moreover, 1α,25(OH)_2_D_3_ increases autophagosomes and LC3 protein levels in PGCs, while the inhibitor of ROS reverses this effect. Recent studies have shown ROS can initiate the formation of autophagosomes and autophagic degradation [47], and autophagy, in contrast, serves to reduce oxidative damage and ROS levels by removing protein aggregates and organelles such as mitochondria [48]. In general, there is a relative balance of ROS produced by mitochondria in the cell. Once the balance is disrupted, the cell gets rid of the excess mitochondria. Our results showed that the account of mitochondria was increased, which is the primary reason for the ROS increase in PGCs treated with 1α,25(OH)_2_D_3_. Together, these results indicated that 1α,25(OH)_2_D_3_-induced ROS promotes autophagy in PGCs.

Accumulating evidence suggests that ROS can induce autophagy through various mechanisms. Here, we detected the gene expression and protein level of BNIP3 and PINK1 in PGCs treated with 1α,25(OH)_2_D_3_; the results showed that 1α,25(OH)_2_D_3_-induced PGC autophagy is mainly mitophagy caused by ROS. It has been confirmed that various mechanisms were involved in ROS-induced autophagy, such as ROS–NRF2–P62 [49], ROS–HIF1–BNIP3/NIX [50], and ROS–TIGAR [51]. BNIP3 is a receptor for mitophagy, an autophagy process that eliminates excess or damaged mitochondria. Previous studies have shown that BNIP3 plays a vital role in PINK1 localization to the outer mitochondrial membrane and proteolysis [52]. Our results showed that 1α,25(OH)_2_D_3_ could induce the expression of BNIP3 and PINK1 mRNA and protein levels, and NAC could reverse the effect of 1α,25(OH)_2_D_3_ by reducing intracellular ROS. It has been reported that BNIP3 promotes autophagic cell death in response to hypoxia; however, Bellot et al. identified autophagy induced by BNIP3 in response to hypoxia as a mechanism to promote tumor cell survival [53]. Although autophagy activation is critical, autophagy is not always beneficial for cell survival or death. The results of this study indicate that 1α,25(OH)_2_D_3_ promotes cell survival and activates PGC autophagy, which is related to 1α,25(OH)_2_D_3_-induced ROS. These results suggest that 1α,25(OH)_2_D_3_-induced ROS induces PGC mitophagy via the BNIP3-PINK1 pathway.

The reproductive performance of sows is an essential factor affecting the economic benefits of the pig industry. Improving the reproductive performance of sows is also one of the goals pursued by breeders and pork producers. The number of ovulations in sows depends on the number of follicles initially collected and the number of terminal atresias. Autophagy and apoptosis of GCs in follicles are closely related to follicular atresia [4]. Follicle growth and development is a complex biological process which is regulated by many factors, including various steroid hormones, metabolic enzymes, and local growth factors [54]. The close relationship between the changes in steroid hormones and synthetic enzymes and GC autophagy is rarely reported. Based on this, the effects of 1α,25(OH)_2_D_3_ on the synthesis of E_2_ and P_4_, proliferation, and autophagy were investigated in PGCs, and the specific mechanisms were explored to provide a theoretical basis for improving the reproductive efficiency of animals. These results provide new insights into the ability of 1α,25(OH)_2_D_3_ to regulate the biological function of GCs and follicular development, which may have reference significance for the study of the reproductive performance of pigs.

## 4. Materials and Methods

### 4.1. Cell Culture

Granulosa cells were isolated and cultured using the method described by Jiang et al. [55]. Porcine ovaries were collected from local slaughterhouses from commercial pigs aged about 1 year, independent of the stage of the estrus cycle. About 20–30 ovaries were collected and transported to the laboratory in saline with penicillin (100 IU/mL) (Gibco-BRL, Gaithersburg, MD, USA) and streptomycin (100 mg/mL) (Gibco-BRL) within 1 h. The ovaries were washed twice with 75% alcohol and then 2–3 times with saline buffer (0.9%, PH < 7) (37 °C). For each replicate, at least 20 ovaries were collected to obtain sufficient GCs from follicles. Medium-sized follicles (3–5 mm) were selected, and a 10 mL syringe was used to aspirate follicular fluid with GCs. Then, the cell suspension was filtered through a 40 μm cell filter, and the mixture was centrifuged at 800× *g* for 5 min to remove the follicular fluid. Granulosa cell pellets were resuspended in DMEM/F12 (Gibco-BRL) medium. Cell viability was determined by trypan blue exclusion (Solarbio Technology Co., LTD, Beijing, China). For cell culture, cells were diluted with DMEM/F12 and seeded in tissue culture plates at a specific density (5 × 10^3^ cells/well for 96-Well, 1 × 10^6^ cells/well for 24-well). DMEM/F12 of diluted cells contained the following substances: 4 ng/mL sodium selenite, 10 mM sodium bicarbonate, 0.1% bovine serum albumin (BSA), 100 U/mL of penicillin, 100 μg/mL streptomycin, 1 mmol/L non-essential amino acid mix, 2.5 μg/mL transferrin, 10 ng/mL bovine insulin, 10^−7^ M androstenedione, and 1 ng/mL bovine FSH (Bioniche Inc., Belleville, ON, Canada). The cells were cultured at 37 °C in 5% CO_2_ and 95% air. After 24 h, the medium was replaced, and then the treatment was carried out. At least three independent replicates were performed.

### 4.2. Cell Viability Assay

The CCK-8 (Beyotime Biological Technology Co Ltd., Shanghai, China) proliferation assay was used to evaluate the cell viability of PGCs. The cells were seeded at a specific density (5 × 10^3^ cells/well) into 96-well plates, and then the cells were treated with 1α,25(OH)_2_D_3_ (1 nM, 10 nM, 100 nM, 200 nM; 12 h, 24 h, 36 h, 48 h) (Solarbio), chloroquine (10 μM, 24 h) (Sigma-Aldrich, St. Louis, MO, USA), or N-acetylcysteine (2 mM, 4 mM, 8 mM; 24 h) (Beyotime) at different doses and times. After various treatments, 10 μL of CCK-8 solution was added to each well and placed at 37 °C for 3 h. The treated wells had cells, culture medium, CCK-8 solution, and drugs. The control (untreated) wells had cells, culture medium, and CCK-8 solution. The blank wells had culture medium and CCK-8 solution. The absorbance of each well was measured at a wavelength of 450 nm. For each group of 4–6 wells, the average of their optical density (OD) was calculated as follows: cell viability (%) = [treated wells OD − blank wells OD] / [control wells OD − blank wells OD] × 100%. At least three independent replicates were performed.

### 4.3. DNA and RNA Extraction and Quantitative Real-Time PCR

Approximately 1 × 10^6^ cells/well were seeded into 24-well plates. After being attached, PGCs were treated with 10 nM 1α,25(OH)_2_D_3_, chloroquine (10 μM), and NAC (4 mM) for 24 h. DNA was extracted using Universal Genomic DNA Purification Mini Spin Kit (Beyotime). RNA was extracted using SimplyP Total RNA Extraction Kit (Hangzhou Bioer Technology Co Ltd., Hangzhou, China) according to the manufacturer’s protocol. Briefly, cells were collected, and 300 μL RIPA was added. Then, the binding solution was added, the mixture was transferred to the purification column and centrifuged at 12,000× *g* for 30 s, and the liquid in the tube was discarded. After that, washing liquid was added and centrifuged at 12,000× *g* for 30 s. The RNA purification column was transferred to the RNA eluent tube, and 40 μL eluent was added and centrifuged at the highest speed (14,000–16,000 g) for 30 s. The purified RNA was obtained. The process of DNA extraction is similar. Related solvents were added successively and the purified DNA was extracted by centrifugation. The concentration of DNA and RNA was measured by microvolume spectrophotometer (Thermo Fisher Scientific, Waltham, MA, USA). The A260/A280 ratio ranges from 1.9 to 2.0. RNA (500 ng) was reverse transcribed into cDNA according to the FastKing cDNA First Strand Synthesis Kit instructions (Tiangen Biochemical Technology Co Ltd., Beijing, China). Then, the mRNA expression level of genes was measured using ChamQ SYBR qPCR Master Mix (Vazyme, Nanjing, China) in a StepOnePlus real-time PCR System (Applied Biosystems, Foster, CA, USA). The reaction volume was 20 μL: 10 μL 2×ChamQ SYBR qPCR Master Mix, 0.4 μL F-primer (10 μM), 0.4 μL R-primer (10 μM), 5 μL template DNA/cDNA, 4.2 μL ddH_2_O. The primer sequences are shown in Table 1. The common thermal cycling parameters of RT-PCR are as follows: pre-denaturation: 95 °C for 3 min. Cycle reaction: 95 °C for 10 s; 60 °C for 30 s; 40 cycles. Melting-curve: 95 °C for 15 s; 60 °C for 1 min; 95 °C for 15 s. Melting-curve analyses were performed to verify product identity. Target gene expression was quantified relative to GADPH expression. The 2^−∆∆Ct^ method was used to calculate the relative gene expression. All samples were run in triplicate.

### 4.4. Western Blot Analysis

Approximately 1 × 10^6^ cells/well were seeded into 24-well plates. After being attached, PGCs were treated with 10 nM 1α,25(OH)_2_D_3_, chloroquine (10 μM), and NAC (4 mM) for 24 h. Protein was extracted with RIPA buffer (Beyotime) following the manufacturer’s protocol and quantified using the bicinchoninic acid (BCA) protein assay kit (Beyotime). Briefly, after the PGCs were treated, the medium was removed and the cells were washed twice with PBS. RIPA buffer was treated with the cells for 30 min (on ice). The supernatant was collected by centrifugation at 12,000 r/min. The BCA working solution was prepared, the protein standard concentration was 0.5mg/mL, and the protein standard was added to the 96-well plate according to the following amounts: 0, 1, 2, 4, 8, 12, 16, 20 μL. Appropriate volume protein samples were added to standard wells (refill to 20 μL per well). BCA working solution (200 μL) was added to each well for 30 min at 37 °C. The wavelength between A562–595 nm was determined, and the protein concentration was calculated according to the standard curve. The proteins were adjusted to the same concentration by sample buffer. Cytosolic protein (20 μg) was subjected to 12% SDS-PAGE and transferred to polyvinylidene difluoride (PVDF) membranes (Beyotime) in a Bio-Rad wet Blot Transfer Cell apparatus (transfer buffer: 39 mM glycine, 48 mM Tris-base, 1% SDS, 20% methanol, pH 8.3). The obtained membranes were blocked in QuickBlockTm Western’s blocking buffer (Beyotime) for 1 h at room temperature. Membranes were washed in TBST (150 mM NaCl, 2 mM KCl, 25 mM Tris, 0.05% Tween20, pH 7.4) and incubated with primary antibodies: β-actin (ACTB) (Mouse polyclonal to ACTB, 1:5000 dilution) (Proteintech Group, Inc., Chicago, IL, USA), LC3 (Rabbit polyclonal to LC3I/II, 1:2000 dilution) (Abcam, Cambridge, UK), BNIP3 (Rabbit polyclonal to BNIP3, 1:1000 dilution) (Abcam), PINK1 (Rabbit polyclonal to PINK1, 1:1000 dilution) (Cell Signaling Technology, Danvers, MA, USA), STAR (Rabbit polyclonal to STAR, 1:500 dilution) (Cell Signaling Technology), P62 (Rabbit polyclonal to P62, 1:10,000 dilution) (Servicebio, Wuhan, China) in QuickBlock™ Primary Antibody Dilution Buffer (Beyotime) overnight at 4 ℃. Membranes were then washed and labeled for 2 h at room temperature with anti-rabbit HRP-conjugated IgG goat (1:4000 dilution) or anti-mouse HRP-conjugated IgG (1:4000 dilution) (Sungene Biotechnology, Tianjin, China) diluted in QuickBlock™ Secondary Antibody Dilution Buffer (Beyotime). Finally, membranes were washed in TBST, and the protein bands were visualized with a chemical luminous imaging system (Millipore, Billerica, MA, USA).

### 4.5. Analysis of Steroid Hormone Production

Approximately 1 × 10^6^ cells/well were seeded into 24-well plates. After being attached, PGCs were treated with 10 nM 1α,25(OH)_2_D_3_ in the presence/absence of chloroquine (10 μM) for 24 h. E_2_ and P_4_ in the medium were measured by the specific ELISA kit (Ruixin Biological Technology Co., Ltd., Quanzhou, China) following the manufacturer’s protocol. Briefly, medium in the co-culture system was collected by centrifuging at 1000 g for 10 min at 4 °C, and the liquid supernatant was used for steroid assays. Each sample was measured 5 times and averaged. The inter- and intra-assay CVs of E_2_ were 5.6% and 3.4%, and that of P_4_ were 6.9% and 7.9%. The minimum detected concentrations of E_2_ and P_4_ were 4.8 pg/mL and 1.45 ng/mL, respectively.

### 4.6. Measurement of Reactive Oxygen Species

ROS generation was detected by DCFH-DA (Beyotime). Fluorescence intensity was measured by a fluorescence microplate reader, fluorescence microscope, or flow cytometry.

Fluorescence microplate reader: approximately 1 × 10^6^ cells/well were seeded into 24-well plates. After being attached, PGCs were treated with 10 nM 1α,25(OH)_2_D_3_ in the presence/absence of N-acetylcysteine (2 mM, 4 mM, 8 mM) for 24 h. After treatments, the cells were co-cultured with 10 μM DCFH-DA for 30 min, the residual DCFH-DA was removed, and the cells were washed with PBS 3 times. The intracellular fluorescence was read by a fluorescence microplate reader at an excitation/emission wavelength of 488/525 nm. Each sample was measured 3 times and averaged.

Fluorescence microscope: approximately 1 × 10^6^ cells/well were seeded into 24-well plates. After being attached, PGCs were treated with 10 nM 1α,25(OH)_2_D_3_ in the presence/absence of N-acetylcysteine (4 mM) for 24 h. Then the cells were stained with DCFH-DA as described above. Cells were observed under a fluorescence microscope, and fluorescence images were obtained. LED intensity, integration time and camera gain were fixed during taking pictures (Olympus Corporation, Tokyo, Japan). Image J software was used to process the images, and the mean fluorescence values of different groups were calculated.

Flow cytometry: approximately 1 × 10^6^ cells/well were seeded into 24-well plates. After being attached, PGCs were treated with 10 nM 1α,25(OH)_2_D_3_ for 24 h. The cells were stained with DCFH-DA as described above. After staining, cells in each group were collected (about 1 × 10^6^ cells/mL), and ROS was detected by flow cytometry (BD FACSAria™ III) (Becton Dickinson, Franklin Lakes, NJ, USA) within 30 min. The excitation light was 488 nm, and the emission light was 525 nm. Fluorescence was detected by the FL1 channel. Samples were acquired on a flow cytometer using a stop condition of 10,000 events on the gate of interest. Using the flow cytometry software, dot plots of FSC (on the *X*-axis) and SSC (on the *Y*-axis) were opened, and a gate was drawn around the cells of interest. In the experiment, untreated normal cells were set as the control group and the gate position was developed according to the two-parameter scatter plot of the control group. Data were analyzed using the FlowJO software.

### 4.7. Detection of Mitochondrial Abundance

Approximately 1 × 10^6^ cells/well were seeded into 24-well plates. After the cells were attached, they were treated with 10 nM 1α,25(OH)_2_D_3_ for 24 h. Cells were stained with Mito-Tracker Green (Beyotime) according to the manufacturer’s instructions. After removing the cell culture medium, the cells were incubated with prepared Mito-Tracker Green working solution for 30 min at 37 °C. Then the cells were washed with PBS 3 times. The cells were treated with an anti-fluorescence quenched sealing solution and then observed under a fluorescence microscope (Olympus Corporation). LED intensity, integration time, and camera gain were fixed during picture-taking. Image J software was used to process the images, and the mean fluorescence values of different groups were calculated.

### 4.8. Mitochondrial Membrane Potential Detection

Approximately 1 × 10^6^ cells/well were seeded into 24-well plates. After the cells were attached, they were treated with 10 nM 1α,25(OH)_2_D_3_ for 24 h. Then, the cells were treated with a mitochondrial membrane potential detection kit (JC-1) (Solarbio) following the manufacturer’s protocol. Briefly, JC-1 staining working solution was added to each well, thoroughly mixed, and then the cells were incubated for 20 min at 37 °C in a cell incubator. The cells were washed 2–3 times with PBS. After staining, cells in each group were collected (about 1 × 10^6^ cells/mL), and the JC-1 signal was visualized by flow cytometry (excitation: 488 nm; emission: 530 nm) (BD FACSAria™ III) within 30 min. Green fluorescence was detected through FL1 channel, and red fluorescence was detected through FL2 channel. Samples were acquired on a flow cytometer using a stop condition of 10,000 events on the gate of interest. Using the flow cytometry software, dot plots of FSC (on the *X*-axis) and SSC (on the *Y*-axis) were opened, and a gate was drawn around the cells of interest. In the experiment, untreated normal cells were set as the control group, and the gate position was developed according to the two-parameter scatter plot of the control group. Data were analyzed using the FlowJO software.

### 4.9. Double Staining with MDC and DAPI

Approximately 1 × 10^6^ cells/well were seeded in 24-well plates. After being attached, PGCs were treated with 10 nM 1α,25(OH)_2_D_3_, chloroquine (10 μM), and NAC (4 mM) for 24 h. MDC (monodansylcadaverine) was used as a tracer of autophagic vesicles. The autophagosomes are marked as clear green dots under the fluorescence microscope. After treatment, the cells were treated with MDC (0.05 mM) (Kaiji Biotechnology Co., Ltd., Nanjing, China) and DAPI (1 μg/mL) (4′,6-diamidino-2-phenylindole) (Solarbio) following the manufacturer’s protocol. Briefly, the cells were grown with MDC and DAPI at 37 °C for 15 min and fixed immediately with paraformaldehyde (4%) in PBS for 20 min, then observed under a fluorescence microscope (Olympus Corporation). LED intensity, integration time, and camera gain were fixed while taking pictures. Image J software was used to process the images, a total of 200 cells in each sample were analyzed, and the percentage of cells with green spots indicates the percentage of autophagy.

### 4.10. Measurement of Superoxide Dismutase and Glutathione Peroxidase

Approximately 1 × 10^6^ cells/well were seeded in 24-well plates. After being attached, PGCs were treated with 10 nM 1α,25(OH)_2_D_3_ for 24 h. The activity of intracellular SOD and GPX were measured in PGCs using kits (Nanjing Jiancheng Bioengineering Research Institute Co., Ltd., Nanjing, China) for scientific research following the manufacturer’s protocol. Briefly, after the cells were washed twice with PBS, the cells were carefully scraped off with a cell scraper, and the cell mixture was centrifuged at 1000× *g* for 10 min, and then the supernatant was discarded. Protein concentration was determined with the BCA assay kit (Beyotime). The results were detected by a visible spectrophotometer (550 nm wavelength) (Thermo Fisher Scientific). Each sample was measured 5 times, and the average of the results was taken. The inter- and intra- assay CVs averaged 7.8% and 6.5%, respectively.

### 4.11. Statistical Analysis

Independent *t*-tests were used to evaluate the significance of the results between groups. Statistical significance was determined by ANOVA followed by post hoc tests. The Tukey–Kramer HSD test was used to analyze the differences between the means (GraphPad Prism version 9.0, GraphPad Software Inc., San Diego, CA, USA). *p*-values < 0.05 were considered statistically significant. All data are presented as the mean ± SM of 3 or more repeated observations from at least 3 independent experiments.

## 5. Conclusions

In summary, these results demonstrate for the first time that 1α,25(OH)_2_D_3_ induces mitophagy through the ROS-BNIP3-PINK1 signaling pathway, which promotes the proliferation and maintains the function of PGCs. Our results provide important information for determining the role of 1α,25(OH)_2_D_3_ during ovarian follicular development.

## Figures and Tables

**Figure 1 ijms-24-04364-f001:**
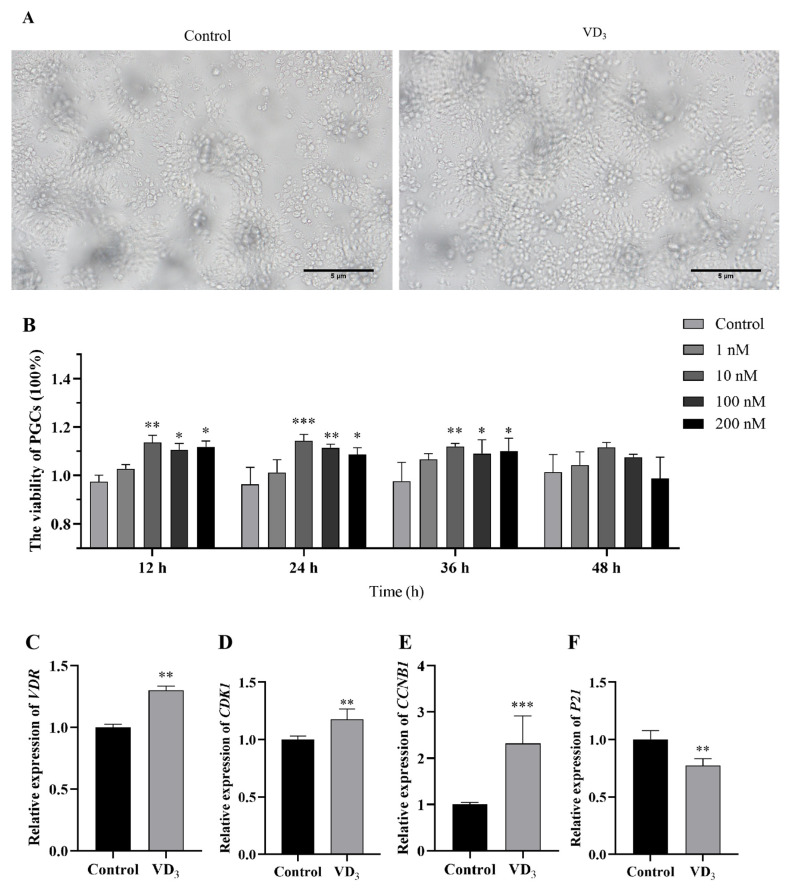
Effects of 1α,25(OH)_2_D_3_ on PGC viability. (**A**) The proliferation of PGCs under a light microscope after 24 h treatment. (**B**) The relative viability of PGCs treated with the different doses of 1α,25(OH)_2_D_3_ for different times (two-way ANOVA and post hoc Student’s *t* test). Relative expression of *VDR* (**C**), *CDK1* (**D**), *CCNB1* (**E**), and *P21* gene (**F**) in PGCs. Bar = 5 μm. VD_3_: 1α,25(OH)_2_D_3_. Data are means ± SE of three independent replicates (* *p* < 0.05, ** *p* < 0.01, *** *p* < 0.001).

**Figure 2 ijms-24-04364-f002:**
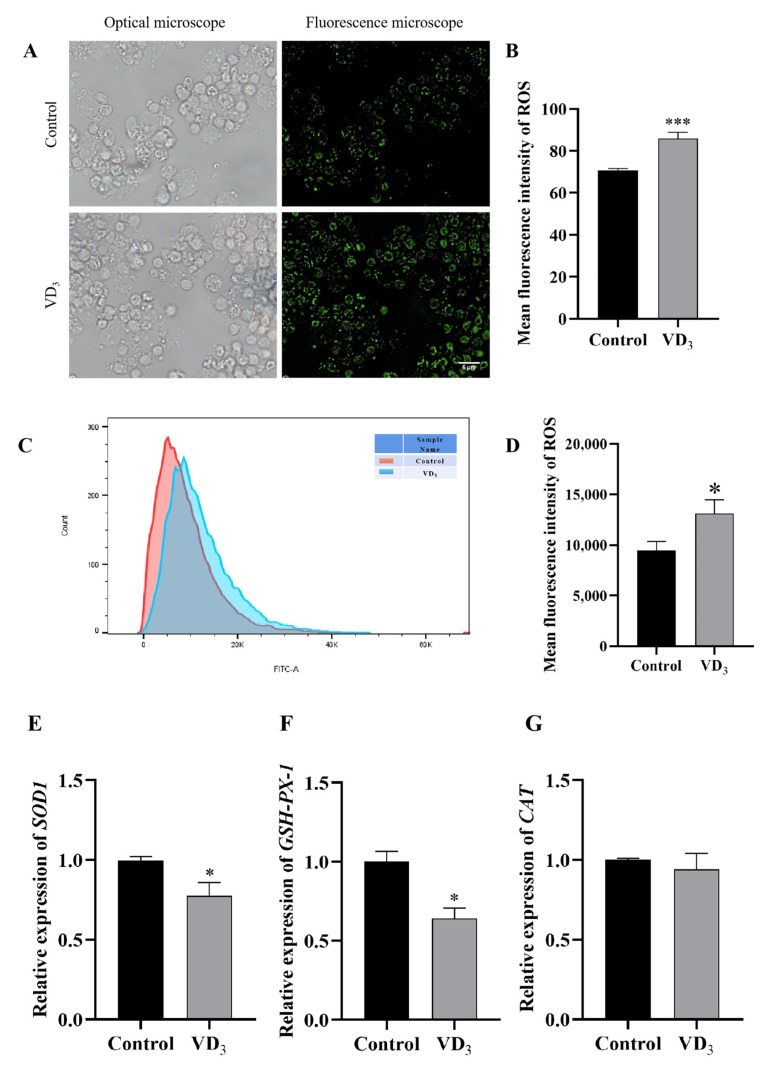
1α,25(OH)_2_D_3_ increases intracellular ROS in PGCs. (**A**) The intracellular ROS was measured by fluorescence microscope (green fluorescence intensity represents ROS levels), and the ROS content in PGCs was evaluated (**B**). (**C**) The intracellular ROS was measured by flow cytometry, and the ROS content in PGCs was evaluated (**D**). Relative gene expression of *SOD1* (**E**), *GSH-PX-1* (**F**), and *CAT* (**G**) were measured in PGCs. Enzyme activities of SOD (**H**) and GPX (**I**). Bar = 5 μm. VD_3_: 1α,25(OH)_2_D_3_. Data are means ± SE of three independent replicates (* *p* < 0.05, ** *p* < 0.01, *** *p* < 0.001).

**Figure 3 ijms-24-04364-f003:**
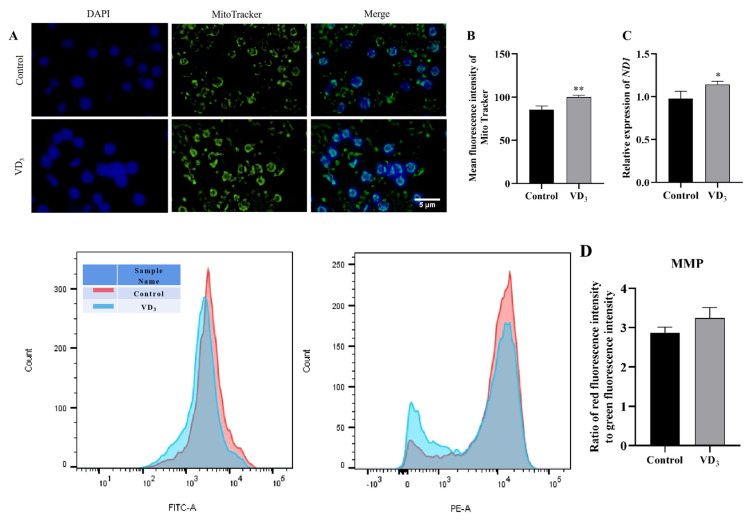
Mitochondria status in the 1α,25(OH)_2_D_3_-treated GCs. (**A**) The Mito-Tracker labeled mitochondria in PGCs, and (**B**) the fluorescence intensity of mitochondria (green) labeled by Mito-Tracker in PGCs. (**C**) Relative gene expression of *ND1* in PGCs. (**D**) The mitochondrial membrane potential of PGCs. Blue fluorescence represented the nucleus. Bar = 5 μm. VD_3_: 1α,25(OH)_2_D_3_. Data are means ± SE of three independent replicates (* *p* < 0.05, ** *p* < 0.01).

**Figure 4 ijms-24-04364-f004:**
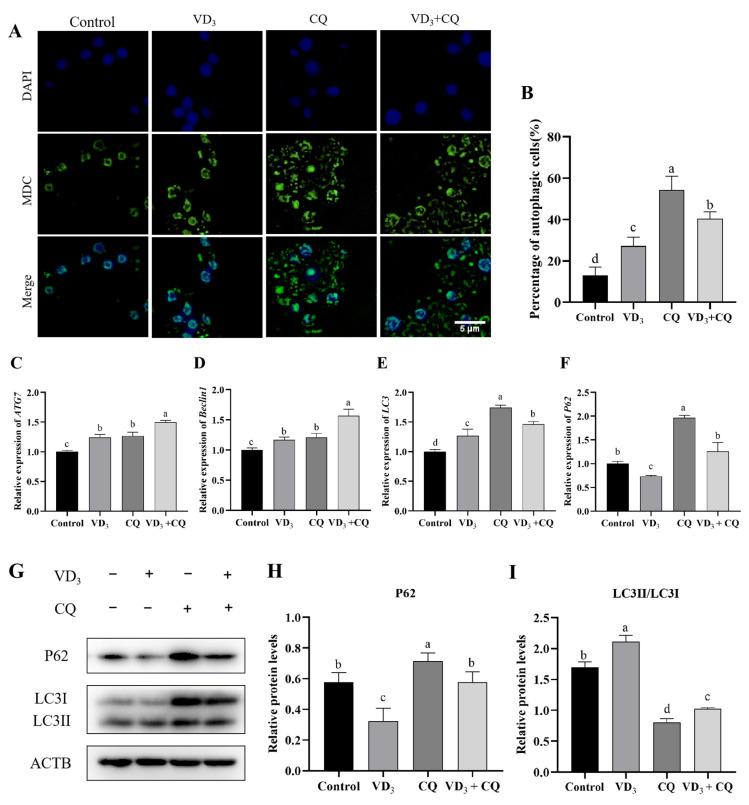
1α,25(OH)_2_D_3_ induces PGC autophagy. (**A**) MDC was used to label autophagic vacuoles (green fluorescence) in PGCs, and the statistical results of the percentage of autophagic PGCs (**B**). The relative gene expression of *ATG7* (**C**), *Beclin1* (**D**), *LC3* (**E**), and *P62* (**F**) in PGCs. Western blotting was used to identify the protein level (**G**), and the protein contents of P62 (**H**) and LC3II/I (**I**) were evaluated in PGCs. Blue fluorescence represented the nucleus. Bar = 5 μm. VD_3_: 1α,25(OH)_2_D_3_. CQ: chloroquine. Different letters on the bar indicate a significant difference. Data are means ± SE of three independent replicates (*p* < 0.05).

**Figure 5 ijms-24-04364-f005:**
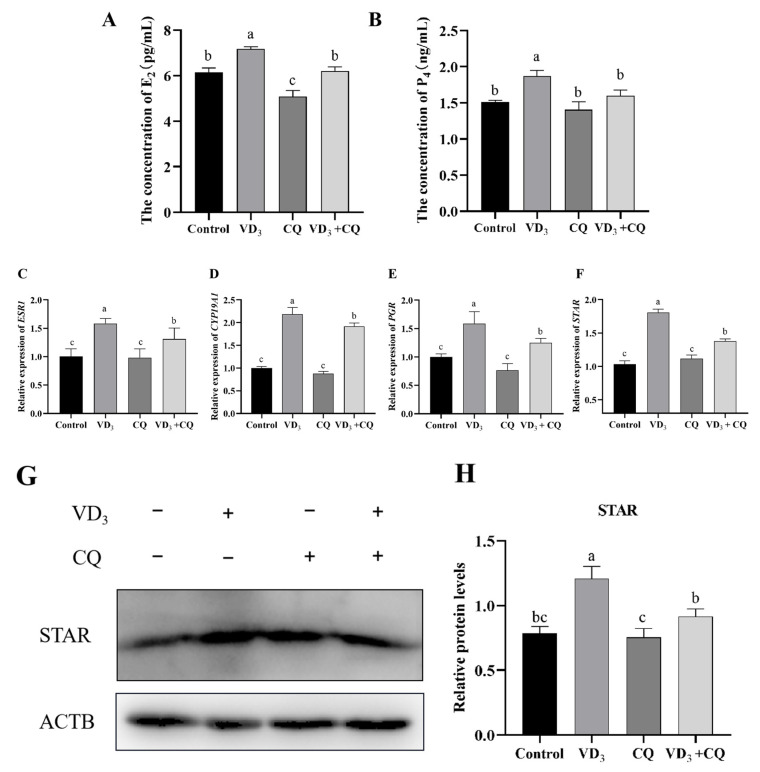
The effects of 1α,25(OH)_2_D_3_ on steroid hormone secretion. (**A**) The content of E_2_ and P_4_ (**B**) production in PGCs. The relative gene expression of *ESR1* (**C**), *CYP19A1* (**D**), *PGR* (**E**), and *STAR* (**F**) in PGCs. Western blotting was used to identify the protein level (**G**), and the protein contents of STAR (**H**) were evaluated in PGCs. VD_3_: 1α,25(OH)_2_D_3_. CQ: chloroquine. Different letters on the bar indicate a significant difference. Data are means ± SE of three independent replicates (*p* < 0.05).

**Figure 6 ijms-24-04364-f006:**
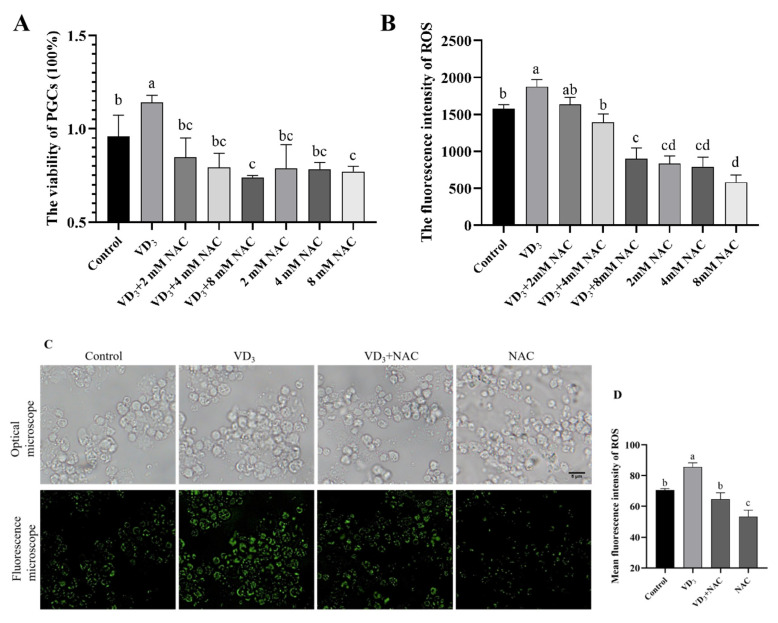
1α,25(OH)_2_D_3_-induced ROS promotes autophagy in PGCs. (**A**) The relative viability of PGCs treated with the different doses of NAC in the presence/absence of 1α,25(OH)_2_D_3_ for 24 h. The intracellular ROS was measured by a multifunctional microplate reader (**B**) and fluorescence microscope (**C**), and the ROS content was evaluated (**D**). MDC-labeled autophagic vacuoles in PGCs (**E**). The fluorescence intensity of autophagic vacuoles was evaluated (**F**). The relative gene expression of *ATG7* (**G**), *Beclin1* (**H**), *LC3* (**I**), and *P62* (**J**) in PGCs. Western blotting was used to identify the protein level (**K**), and the protein contents of LC3II/I (**L**) and P62 (**M**) were evaluated in PGCs. Bar = 5 μm. VD_3_: 1α,25(OH)_2_D_3_. NAC: N-acetylcysteine. Different letters on the bar indicate a significant difference. Data are means ± SE of three independent replicates (*p* < 0.05).

**Figure 7 ijms-24-04364-f007:**
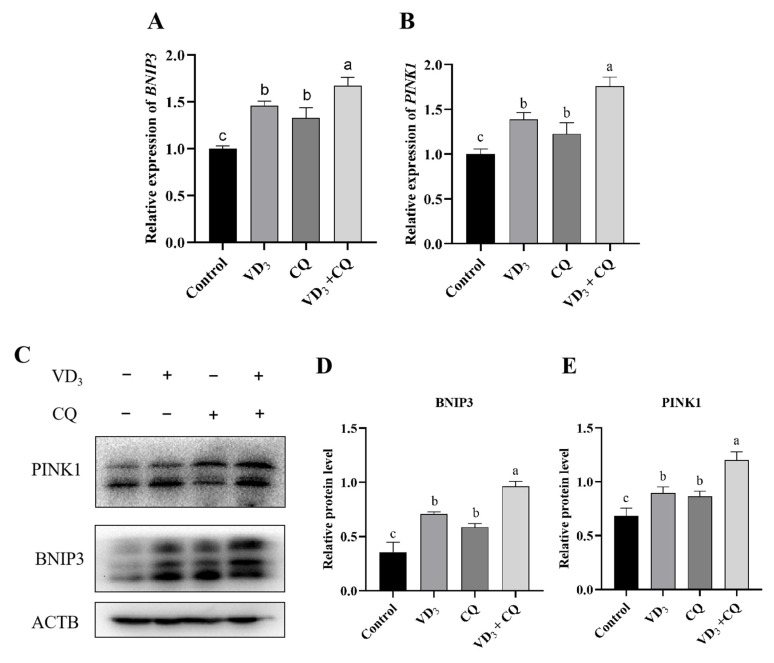
1α,25(OH)_2_D_3_ induces mitophagy in PGCs through the ROS-BNIP3-PINK1 signaling pathway. The relative gene expression of *BNIP3* (**A**,**F**) and *PINK1* (**B**,**G**) in PGCs. Western blotting was used to identify the protein level (**C**,**H**), and the protein contents of BNIP3 (**D**,**I**) and PINK1 (**E**,**J**) were evaluated in PGCs. VD_3_: 1α,25(OH)_2_D_3_. NAC: N-acetylcysteine. Different letters on the bar indicate a significant difference. Data are means ± SE of three independent replicates (*p* < 0.05).

**Table 1 ijms-24-04364-t001:** Porcine-specific Q-PCR primer sequences.

Gene	Primer Sequence (5′-3′)	Genebank No.	Size (bp)
*GAPDH*	F: GGACTCATGACCACGGTCCATR: TCAGATCCACAACCGACACGT	XM_021091114.1	220
*ATG7*	F: AGATTGCCTGGTGGGTGGTR: GGGTGATGCTGGAGGAGTTG	XM_021068402.1	140
*BECN1*	F: AGGAGCTGCCGTTGTACTGTR: CACTGCCTCCTGTGTCTTCA	XM_013980932.2	189
*STAR*	F: CGTCGGAGCTCTCTTCTTGGR: CCTCCTGGTTGCTGAGGATG	NM_213755.2	124
*PGR*	F: GTCGCCAGCTATAGGGAACCR: AAGCAATTACTGGCGGCTCT	NM_001166488.1	178
*SQSTM1*	F: AAGCTGAGACATGGGCACTTR: ACACTCTCCCCTACGTTCTTG	XM_003123639.4	173
*LC3*	F: GCCTCTCAGGAGACTTTCGGR: GAGCTCCGTTTTTCTGCGTG	NM_001190290.1	214
*VDR*	F: CCGGACCAGAGTCCTTTTGGR: ATGCGGCAGTCTCCATTGAA	XM_021091108.1	298
*ESR1*	F: ATGGCCATGGAATCTGCCAAR: CCCCTTTCATCATGCCCACT	XM_021083075.1	241
*CYP19A1*	F: GTGGACGTGTTGACCCTCATR: GGCACTTTCATCCAAGGGGA	NM_214431.2	84
*ND1*	F: AGCCATGTCAAGCCTAGCAGR: ATGGCTAGGGGTCAGGATGT	NC_012095.1	231
*SOD1*	F: GCGAGTCATGGCGACGAAR: CCTTTGGCCCACCATGTTTT	NM_001190422.1	233
*GSH-PX1*	F: CTAGCAGTGCCTAGAGTGCCR: CGCCCATCTCAGGGGATTTT	NM_214201.1	142
*CAT*	F: CCTGCAACGTTCTGTAAGGCR: GCTTCATCTGGTCACTGGCT	NM_214301.2	72
*BNIP3*	F: TCTGCAGCCTTGAGTTAGGCR: CGGATCTGTAGCCTGGGTTC	XM_003359404.4	181
*PINK1*	F: AGACCTGCAGTTGTTAGCCCR: CCACCCCAGGCCTCATTAAG	XM_021095478.1	261

## Data Availability

Not applicable.

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
