# Peer review of "1α,25(OH)_2_D_3_ Promotes the Autophagy of Porcine Ovarian Granulosa Cells as a Protective Mechanism against ROS through the BNIP3/PINK1 Pathway"

_ijms, 2023, doi:10.3390/ijms24054364_

Round 1

Reviewer 1 Report

Comments about the manuscript:

“1α,25(OH)2D3 promotes the autophagy of porcine ovarian granulosa cells as a protective mechanism against ROS through BNIP3/PINK1 pathway”

Vitamin D deficiency, important for livestock, is widespread. This vitamin would have an effect on reproduction, among other things, but this point remains to be clarified. The purpose of the work presented here was to determine the role of 1,25-dihydroxy vitamin D3 (1α,25(OH)2D3) on granulosa in sows. For this, the authors used an in vitro ovary model subjected to different treatments. They were thus able to show that vitamin D3 regulated granulosa cell autophagy and ROS content.

This article brings useful elements and could be published after some improvements. Here are some remarks.

Page 2, lines 87-88. “Porcine ovaries were collected from local slaughterhouses from commercial pigs aged about one year, independent of the stage of the estrus cycle.”: As this is work involving material of animal origin, it is necessary to specify whether the study has been authorized by a university ethics committee or another and to provide the references of the authorizations, agreement.

Page 2, lines 91-92. “with saline (37°C)”: is it a saline buffer? What was its composition and pH?

Page 3, lines 113-114. “After various treatments, 10 μL of CCK-8 solution was added to each well”: is it not: “After various treatments, 10 μL of CCK-8 solution were added to each well”?

Page 3, lines 126-127. “according to the manufacturer’s protocol.” seems insufficient to me: it would be useful to briefly recall the protocol (as it has been written below for other protocols).

Page 4, lines 146-147: same: “following the manufacturer’s protocol” seems insufficient to me: it would be useful to briefly recall the protocol (as it has been written below for other protocols).

Page 8, figure 1, legend: “(A) The proliferation of PGCs”: please specify at what time of the treatment?

Page 19 line 572. “The number of eggs ovulated by sows”: I disagree with the word "egg" here. the egg is the cell obtained after the fertilization of the ovum with the spermatozoon. It is an oocyte.

Reviewer 2 Report

This is an interesting manuscript by Wang et al., which shows that 1α,25(OH)2D3 promotes protective autophagy by ROS induction through BNIP3/PINK1 pathway in porcine ovarian granulosa cells. The applied science behind this research article is pig farm production, and their fecundity plays a crucial role in the benefit of pig farms. However, there are important parameters that need to be addressed before the manuscript is publishable.

1. The author has not shown how VD3 affected the reproductive physiology of porcine. For example, no. of oocytes ovulated and their ability to be fertilized. This is important to determine healthy embryos.

2. Moreover, autophagy and apoptosis often work in accordance, so it will be interesting to see the status of autophagic proteins in presence of pan-caspase inhibitors (Z-VAD-FMK)

3. Since there is ROS production, it is important to show how autophagic machinery attenuated the DNA damage.

Round 2

Reviewer 2 Report

Though no new experiments have been incorporated, the authors have provided sufficient literature to support the raised queries. Hence, the decision is to "Accept".